# Unraveling the Molecular Mechanisms of Tomatoes’ Defense against *Botrytis cinerea*: Insights from Transcriptome Analysis of Micro-Tom and Regular Tomato Varieties

**DOI:** 10.3390/plants12162965

**Published:** 2023-08-16

**Authors:** Shifu Tian, Bojing Liu, Yanan Shen, Shasha Cao, Yinyan Lai, Guodong Lu, Zonghua Wang, Airong Wang

**Affiliations:** 1State Key Laboratory of Ecological Pest Control for Fujian and Taiwan Crops, Fujian Agriculture and Forestry University, Fuzhou 350002, China; tianshifu128199@163.com (S.T.); shenyanan0925@163.com (Y.S.); presence_css@163.com (S.C.); m18022812145@163.com (Y.L.); lgd@fafu.edu.cn (G.L.); 2Haixia Institute of Science and Technology, Fujian Agriculture and Forestry University, Fuzhou 350002, China; 3College of Resources and Environment, Fujian Agriculture and Forestry University, Fuzhou 350002, China; liubojing0116@163.com; 4Institute of Oceanography, Minjiang University, Fuzhou 350108, China; 5Fujian Key Laboratory for Monitoring and Integrated Management of Crop Pests, Fuzhou 350003, China

**Keywords:** tomato, Micro-Tom, *B. cinerea*, transcriptome sequencing, Ca^2+^, transcription factors

## Abstract

*Botrytis cinerea* is a devastating fungal pathogen that causes severe economic losses in global tomato cultivation. Understanding the molecular mechanisms driving tomatoes’ response to this pathogen is crucial for developing effective strategies to counter it. Although the Micro-Tom (MT) cultivar has been used as a model, its stage-specific response to *B. cinerea* remains poorly understood. In this study, we examined the response of the MT and Ailsa Craig (AC) cultivars to *B. cinerea* at different time points (12–48 h post-infection (hpi)). Our results indicated that MT exhibited a stronger resistant phenotype at 18–24 hpi but became more susceptible to *B. cinerea* later (26–48 hpi) compared to AC. Transcriptome analysis revealed differential gene expression between MT at 24 hpi and AC at 22 hpi, with MT showing a greater number of differentially expressed genes (DEGs). Pathway and functional annotation analysis revealed significant differential gene expression in processes related to metabolism, biological regulation, detoxification, photosynthesis, and carbon metabolism, as well as some immune system-related genes. MT demonstrated an increased reliance on Ca^2+^ pathway-related proteins, such as CNGCs, CDPKs, and CaMCMLs, to resist *B. cinerea* invasion. *B. cinerea* infection induced the activation of PTI, ETI, and SA signaling pathways, involving the modulation of various genes such as FLS2, BAK1, CERK1, RPM, SGT1, and EDS1. Furthermore, transcription factors such as WRKY, MYB, NAC, and AUX/IAA families played crucial regulatory roles in tomatoes’ defense against *B. cinerea*. These findings provide valuable insights into the molecular mechanisms underlying tomatoes’ defense against *B. cinerea* and offer potential strategies to enhance plant resistance.

## 1. Introduction

Tomatoes (*Solanum lycopersicum*) are an essential vegetable crop and a key model plant in the study of pulp biology. It is the cornerstone of biological research and genetic improvement of all nightshade crops, such as potatoes, peppers, and eggplants [1,2] (pp. 391–394, pp. 355–366). However, continuous planting of tomatoes year after year puts the crop at serious risk of disease, especially from tomato gray mold disease [3] (pp. 2178–2192). Gray mold is a necrotic disease caused by the fungal pathogen *Botrytis cinerea*, which is capable of infecting over 200 plant species worldwide, making it a significant pathogen that is extensively studied [4,5] (p. 334, pp. 613–622). Each year, at least $10 billion is lost globally due to *B. cinerea*, making it the most extensively studied necrotrophic pathogenic fungus [6] (pp. 89–97). Necrotrophic pathogens typically damage plant cells by secreting toxic compounds, lyases, and other pathogenic factors that can inhibit host defenses [7] (p. 594743).

Plants have developed sophisticated and effective systems to recognize and protect against microbial invasion, including innate immunity known as pathogen-associated molecular pattern (PAMP)-triggered immunity (PTI) [8] (pp. 10–16) and effector-triggered immunity (ETI) [9,10,11] (p. e1003313, pp. 1049–1058, p. e1000061). The signaling pathways triggered by PRRs are involved in PTI and NLRs are involved in ETI. The immune system response of tomatoes to *B. cinerea* is primarily regulated by PTI, consisting of receptor proteins, signal transduction pathways, and transcription factors [12,13,14] (pp. 2405–2421, p. 235, pp. 262–274). Receptor proteins play a crucial role by recognizing surface molecules of *B. cinerea*, such as chitin, the main component of the fungal cell wall [15] (pp. 1964–9183). A recent study discovered LeEIX2, a tomato immune receptor molecule that identifies molecular signals released by *B. cinerea* and induces immune responses [16] (pp. 1287–1306). RAR1, an essential immune activation factor, binds with another protein to activate immune responses [17] (p. e970410). Moreover, several other receptor proteins, including LeEix1 [18] (pp. 277–285), Ve1, GLR3.3, and GLR3.5 [19,20,21] (pp. 349–357, pp. 320–332, p. 11217), have been recognized for their role in immune responses against *B. cinerea*. Transcription factors, such as WRKY33 [22,23] (pp. 266–285, pp. 4567–4583), WRKY75 [24] (pp. 1473–1489), and SlNAC1 [25] (pp. 4877–4893), regulate the expression of immune-related genes and are critical for plant defense against *B. cinerea*. Additionally, research has unveiled that the secretion of BcXYG1, one of the exo-1,3-beta-glucanases from *B. cinerea*, act as PAMPs, promoting pathogen virulence while eliciting plant immune responses [26,27,28,29] (pp. 438–456, pp. 260–273, pp. 1914–1932, pp. 2057–2072). *B. cinerea* also produced numerous effectors to enhance its virulence, and some of these effectors can be recognized by plants, triggering effector-triggered immunity (ETI). For instance, BcCrh1 acts as a cytoplasmic effector and an elicitor of plant defense [30] (p. 2166). Recent studies have uncovered shared signaling components in PTI and ETI [31,32] (p. 102030, p. 1527). These complex interactions between the two immune responses are crucial for plants to establish a robust defense mechanism against pathogens. The crosstalk between the PTI and ETI pathways can augment the intensity and duration of immune responses, thereby enhancing plant defense against pathogens.

Calcium (Ca^2+^) is a second messenger for PTI and ETI. It has been observed that exogenous Ca^2+^ can activate the plant’s resistance response against pathogens [33,34] (pp. 595–598, pp. 976–989), and Ca^2+^ also enhances tomatoes’ resistance against *B. cinerea* [35] (pp. 78–82). As a second messenger, Ca^2+^ plays a crucial role in controlling the expression patterns of signaling pathways and is essential for cell survival [36] (p. 100235). Moreover, cyclic nucleotide gated channels (CNGC) [37] (pp. 19–26), specifically the group IVb SlCNGC genes, primarily regulate a broad spectrum of resistance against pathogens by modulating Ca^2+^ signaling and influencing immune responses in plants [36,38] (p. 100235, p. 303).

Among the numerous tomato cultivars, brassinosteroids (BRs)-deficient Micro-Tom (MT) with a mutated DWARF gene [39] (pp. 553–560) has been recognized as a model cultivar for tomato research due to its small size (10–20 cm tall), a rapid life cycle (70–90 d) and easy transformation [40] (pp. 47–55). Currently, a few studies have utilized biochar, other bioformulations, or hormones to induce resistance in tomatoes against gray mold [41,42,43,44] (pp. 150–157, pp. 31–34, pp. 491–501, pp. 1455–1465). However, research on the underlying mechanisms of tomato cultivar resistance to *B. cinerea* is limited. In this study, we compared the development of symptoms on the leaves of MT and Ailsa Craig (AC) at different times after inoculation with *B. cinerea*. Furthermore, transcriptome analysis was used to elucidate the molecular mechanisms underlying this response.

## 2. Results

### 2.1. Different Tomato Varieties Exhibited a Pronounced Response to B. cinerea during the Early Stages of Infection

To assess the response of two distinct tomato cultivars, AC and MT, to *B. cinerea* infection, we inoculated the leaves of 4-week-old plants with a *B. cinerea* conidia solution of 1 × 10^5^ spores/mL. The progression of symptoms was recorded at various time points after inoculation (Figure 1). By 12 h post-inoculation (hpi), no lesions were observed, but a few very small lesions began to appear after 14 hpi. By 18 hpi, all inoculated sites on AC leaves presented minor water-soaked lesions, while MT leaves exhibited almost no lesions. At 22 hpi, light brown lesions characteristic of *B. cinerea* infection were visible on AC leaves, while water-soaked lesions resembling those observed on AC leaves at 20 hpi were observed on MT leaves. At 24 hpi, typical lesions had started to form on MT leaves, but they were smaller than those found on AC leaves. By 36 hpi, the lesions on MT leaves had surpassed those on AC leaves (Figure 1). At 48 hpi, entire MT leaves had rotted, whereas AC leaves had not completely rotted, and the lesions on AC leaves had only slightly increased in size compared to those at 36 hpi. These results suggested that MT was more resistant to early-stage *B. cinerea* infection than AC, while it experienced a faster lesion expansion at later stage of infection.

### 2.2. DEG Analysis of AC and MT Infected Leaves Using RNA-seq

To investigate the differential gene expression profiles of these two cultivars in response to *B. cinerea* infection, we collected samples of inoculated or uninoculated leaves of AC and MT at 24 hpi and 22 hpi, respectively. Total RNA was extracted from these 12 samples and Nanopore sequencing was used to obtain clean data of 2.63 GB per sample. Full-length sequences were identified with primers at both ends of the reads, resulting in 2,415,604 to 3,153,197 sequences per sample (Appendix A). Significantly, the number of sequences in MT samples was remarkably lower than AC samples.

After aligning clean reads to the reference genome, we identified 31,041 annotated genes in at least 1 of the 8 among 12 libraries (Appendix A). The high-quality and reproducibility of the experimental design was confirmed by principal component analysis (PCA) and correlation analysis (Appendix A). The Pearson correlation coefficient (r) was used to measure biological replicate correlation and Appendix A showed a strong correlation among the 3 biological replicate samples with r2 approaching 1.

Using DESeq [45] (p. R106) for samples with biological replicates, differential expression analysis of RNA sequencing data identified a total of 3155 and 5827 differentially expressed genes (DEGs) in AC and MT at 24 and 22 hpi with *B. cinerea*, respectively (Figure 2b). In AC, 1655 genes were upregulated, and 1500 genes were downregulated (Figure 2b and Appendix A). While in MT, the number of DEGs was nearly double that of AC, with 2940 upregulated genes and 2887 downregulated genes (Figure 2a,b). We also found 909 DEGs in inoculated MT samples compared with inoculated AC samples. These significant differences were clearly shown in volcano plot and hierarchical clustering analysis used to compare global gene expression changes (Figure 2c and Appendix A), indicating a widespread activation of defense mechanisms against the pathogen. The MT cultivar exhibited a higher number of DEGs compared to AC, suggesting a more extensive transcriptional reprogramming in response to the infection in MT. These findings indicated that the two cultivars exhibited distinct gene expression patterns and molecular responses in the context of *B. cinerea* infection.

### 2.3. Significant Impact of B. cinerea Infection on Tomatoes’ Biological Pathways

Previous studies have demonstrated the pathogen invasion and plant immune system activation affect carbon metabolism and hormone synthesis [46,47] (p. 10.4172, pp. 239–277). When plants respond to immune challenges, they adjust their carbon metabolism to improve carbon utilization efficiency [48] (pp. 307–318). To explore the impact of *B. cinerea* invasion on the regulation of tomatoes’ biological pathways, we conducted KEGG pathway enrichment analysis to assess the modulation of DEGs. Our analysis revealed significant differences in carbon metabolism, with the highest number of enriched genes observed. In detail, the CKAC-vs-AC group showed 244 enriched genes, the CKMT-vs-MT group had 349 enriched genes, while the AC-vs-MT group showed only 59 enriched genes (Figure 3). In contrast, the pathway for ketone body synthesis and breakdown only showed a minimal number of enriched genes, with 16 and 17 genes in the CKAC-vs-AC group and CKMT-vs-MT group, respectively (Figure 3). These results indicated that the impact of *B. cinerea* infection on tomatoes’ carbon metabolism was significant, whereas the effect on ketone body metabolism was minor, and the C5-branched dibasic pathway within the AC-vs-MT group was negligible.

Activation of the plant immune system can also affect photosynthesis, resulting in reduced photosynthesis efficiency in some cases of pathogen infection [49,50,51,52] (p. 17121, pp. 487–497, pp. 42–49, pp. 531–539). When analyzing the DEGs related to photosynthesis, we observed that 47 and 77 enriched genes in the “photosynthesis-antenna proteins” and “photosynthesis” pathways in the CKAC-vs-AC group, respectively (Figure 3a). In the CKMT-vs-MT group, these pathways contained 48 and 94 enriched genes, while the AC-vs-MT group had only 17 enriched genes in both pathways (Figure 3). This highlighted the potential impact of *B. cinerea* infection on the regulation of genes associated with photosynthesis in tomatoes.

Meanwhile, GO enrichment analysis revealed significant differences in the proportion of DEGs and their functional annotations among three distinct groups: CKAC-vs-AC, CKMT-vs-MT, and AC-vs-MT (Appendix A). Among the total 183,180 annotated genes, CKAC-vs-AC showed 21,645 DEGs (11.8%), while CKMT-vs-MT and AC-vs-MT exhibited 39,672 DEGs (21.7%) and 7945 DEGs (4.3%), respectively. Analysis of DEGs in AC-vs-MT revealed that 42% were associated with cellular components, 21% with molecular functions, and 37% with biological processes. Notably, metabolic processes were prominently represented across all groups, with 2164 and 3765 DEGs identified in CKAC-vs-AC and CKMT-vs-MT, respectively. In contrast, DEGs related to the immune system were limited to 18 and 25 in CKAC-vs-AC and CKMT-vs-MT, respectively. Variations were observed in the number of DEGs related to biological regulation and detoxification. These findings provided valuable insights for further investigation into the functional roles of DEGs in the interplay between tomatoes and *B. cinerea*.

### 2.4. Enhanced Activation of Ca^2+^ Channel-Associated Genes in MT Tomato Variety during Early B. cinerea Attacking

DEG analysis results revealed that during the early stages of *B. cinerea* infection, tomatoes rely significantly on Ca^2+^ channel-associated proteins, including CNGCs, calcium-dependent protein kinases (CDPKs), and calmodulin-like proteins (CaMCMLs), for activation and defense against the pathogen, with a particularly pronounced response observed in the MT variety (Figure 4a,b). Among the DEGs, 11 CNGCs showed significant differential expression, with the majority displaying more pronounced upregulation in MT, such as *Solyc05g050350.1.1* and *Solyc05g050360.2.1*, exhibiting significantly higher upregulation in MT compared to AC. Similarly, 26 CDPKs exhibited significant upregulation during *B. cinerea* infection, with a more prominent differential expression in MT cultivar, exemplified by *Solyc10g074570.1.1* and *Solyc10g079130.1.1*, which showed extremely significant upregulation in MT while displaying relatively mild differential expression in AC cultivar (Figure 5a and Figure 6a). Additionally, respiratory burst oxidase homolog (RBOH) encoding genes such as *Solyc01g099620.2.1* and *Solyc03g117980.2.1* were activated and upregulated during the early stages of *B. cinerea* infection, with *Solyc01g099620.2.1* showing a more prominent upregulation in MT cultivar (Figure 5a and Figure 6a). Furthermore, in response to *B. cinerea* infection, 18 CaMCML genes were activated, with the majority exhibiting more significant upregulation in MT cultivar, including *Solyc01g105630.2.1* and *Solyc02g088090.1.1*, both of which showed extremely significant upregulation in MT compared to AC cultivar (Figure 5a and Figure 6a). These findings underscored the crucial role of these genes in regulating defense against pathogen invasion, particularly in MT cultivar.

### 2.5. Partial Activation of PTI, ETI, and SA Signaling Pathways during Early Infection of Tomatoes by Gray Mold Pathogen

AC and MT cultivars displayed a unique enrichment of plant-pathogen interaction genes, with a total of six genes identified. Among them, two genes, *Solyc05g050350.1.1* and *Solyc02g088560.2.1*, belong to the H subfamily (Eag-related) and are associated with intracellular transport, secretion, and vesicle trafficking. *Solyc01g095100.2.1* is related to WRKY transcription factors. *Solyc05g010670.2.1* exhibits ATPase activity and is involved in post-translational modification of proteins. *Solyc03g005040.1.1* is associated with calcium-binding proteins. *Solyc03g093240.2.1* plays a role in late-stage assembly of the 50S ribosomal subunit and possesses GTPase activity. These six genes were differentially expressed, with four genes upregulated and two genes downregulated (Appendix A). These findings suggested that during the early stage of *B. cinerea* infection in tomato plants, certain intracellular transport, secretion, and vesicle trafficking proteins might be crucial for plants’ immune response, with a partial activation of nuclear immune responses in plant cells.

Furthermore, upon treatment with *B. cinerea*, the expression of certain genes involved in the PTI response showed significant upregulation, indicating their potential role in resisting *B. cinerea* invasion (Figure 4d). Notably, there were differential levels of upregulation among these marker genes between the two tomato cultivars. Specifically, the PRR gene encoding CERK1 demonstrated significantly higher upregulation in MT compared to AC (Figure 6c), while EIX1, FLS2, and BAK1 showed similar upregulation levels in both cultivars (Figure 5d and Figure 6c). Further downstream genes such as MEKK1 and immune marker genes including NHO1, PDF1.2, and PR1 showed highly significant upregulation in MT with significantly higher expression levels than in AC (Figure 5d and Figure 6c).

In addition, several genes involved in the ETI response, such as RPM1 and RPS2, showed downregulation (Figure 4c), while several NBS-LRR genes and PIK1, RIN4, and Pti5 showed upregulation upon *B. cinerea* treatment, with highly significant upregulation observed in MT (Figure 6b). These findings suggested that MT cultivar displayed more pronounced response to *B. cinerea* invasion, indicating that PTI and ETI were partially activated during the early stage of *B. cinerea* invasion in tomato plants, particularly exhibiting stronger reactions in MT.

Moreover, our study revealed that the homologous genes of NPR1, EDS1, and PAD4 in the salicylic acid (SA) signaling pathway were upregulated to various degrees upon *B. cinerea* treatment. While these genes showed slight upregulation in AC, they all exhibited significant upregulation in MT (Figure 6b,d). These results indicated that the SA signaling pathway was activated during the early stage of *B. cinerea* invasion in tomato plants, with MT cultivar demonstrating a more pronounced response in this hormone pathway when encountering *B. cinerea* invasion.

Transcription factors such as WRKYs, MYB, and NACs are widely recognized for their pivotal roles in regulating plant responses to both biotic and abiotic stresses [53,54,55,56] (p. e102067, pp. 1227–1238, pp. 1–25, p. 771). Our analysis revealed the abundance of 18 WRKY transcription factors as the most prevalent TF family in tomatoes’ response to *B. cinerea* infection (Figure 7c). The upregulation of WRKY28 and WRKY29 was observed in response to *B. cinerea* infection, especially in the MT variety (Figure 6c and Figure 7c), suggesting their important role in tomatoes’ defense against *B. cinerea* infection. Furthermore, 14 MYB transcription factors (Figure 7d) and 11 NAC transcription factors (Figure 7b) were all induced to a high level in both tomato cultivars by *B. cinerea* infection, indicating their potential regulatory role in tomatoes’ response to *B. cinerea* infection. Interestingly, we observed a significant upregulation of 11 Auxin/IAA transcription factors upon *B. cinerea* induction, except for *Solyc08g021820.2.1*, which was induced in AC but suppressed in MT (Figure 7a). These findings suggested a potential impact of *B. cinerea* on normal tomato development.

Overall, our study provides important insights into the potential roles of different TF families in regulating tomatoes’ response to *B. cinerea* infection. Further studies are needed to elucidate the specific functions of individual TFs and their potential as targets for improving plant resistance to this pathogen.

## 3. Discussion

Tomatoes are the top-ranked vegetable grown globally [57] (pp. 73–90). And among many cultivars, the dwarf cultivar MT has become a popular model plant for tomato research due to its miniature determinate cultivar and high degree of genetic homogeneity [40] (pp. 47–55). Takahashi et al. (2005) found most of the important tomato pathogens could infect MT and cause typical symptoms, but certain pathogens were restricted on MT [58] (pp. 8–22). MT is more tolerant to the soilborne pathogen *F. oxysporum* f. sp. *lycopersici* (*Fol*) than the tomato cultivar Kremser Perle [59] (p. 595). In this study, we observed that MT was more tolerant to *B. cinerea* than AC at the early stage of infection, as evidenced by the significantly delayed appearance of lesions on MT compared to AC. However, at the late stage, MT was more susceptible to *B. cinerea* than AC as illustrated by the larger lesions compared to AC at 36 hpi (Figure 1). These findings suggest that MT may possess enhanced early resistance to *B. cinerea*, probably the main defense against invasion, but weak resistance to expansion.

*B. cinerea* has been a major contributor to significant yield and quality losses in tomato crops for an extended period, and currently, no disease-resistant varieties are available for production [60,61,62,63,64] (pp. 9923–9932, pp. 6555–6563, pp. 5342–5350, pp. 561–580, pp. 1839–1849). Several transcriptome studies have investigated the interaction mechanisms between *B. cinerea* and tomatoes, as well as the immune molecular mechanisms of tomatoes [65,66,67,68,69,70,71] (p. 911, p. 142, p. 125901, p. 111382, pp. 1–12, p. 3, pp. 505–522). However, it should be noted that different cultivars were used in these studies, and most of them were sequenced using the illumina method. Therefore, further exploration of plant or pathogen factors during their interactions is warranted through transcriptome analysis. Here we inoculated leaves of the two different tomato cultivars (AC and MT) with conidia solutions and collected samples at 22 hpi and 24 hpi, respectively. We conducted transcriptome sequencing analysis using nanopore sequencing technology. Consistent with prior investigations, we observed a significant upregulation of numerous tomato genes that may enhance plant defense against *B. cinerea*. Specifically, RNA-seq analysis detected a total of 33,320 genes from the samples, among which 3155 and 5827 DEGs were identified in the CKAC-vs-AC and CKMT-vs-MT comparisons, respectively (Figure 2a,b). This findings confirm that *B. cinerea* invasion triggers substantial transcriptional reprogramming in tomatoes to defend against pathogenic microorganisms [72] (pp. 3530–3557). Moreover, the response of the two tomato cultivars to *B. cinerea* invasion differed significantly, with MT exhibiting a significantly larger number of DEGs compared to AC, and the extent of upregulation was also much higher in MT.

Our results suggested that *B. cinerea* infection in tomatoes triggered the upregulation of certain genes associated with Ca^2+^ signaling pathways, including CNGC, CDPKs, CaMCMLs, and RBOHs (Figure 4, Figure 5 and Figure 6a). Among them, we validated the upregulation of two CNGCs using qRT-PCR, demonstrating their significant upregulation in the early stages of *B. cinerea* invasion in tomatoes. Notably, the upregulation was much higher in MT than in AC, indicating that MT exhibits a stronger response to *B. cinerea* infection. Previous studies have emphasized the crucial role of calcium ion channels in regulating immune responses during plant pathogen interactions [73,74,75] (pp. 2751–2755, pp. 5391–5404.e5317, pp. 1078–1094). Specifically, CNGC genes and their involvement in regulating intracellular calcium ion flux have been highlighted [75] (pp. 1078–1094). Our qRT-PCR results revealed a significant upregulation of CaMCMLs upon *B. cinerea* induction (Figure 6a), with higher expression levels observed in MT compared to AC. Studies have shown that CaM and CML proteins decode calcium signals by interacting with CaM/CML-binding proteins, particularly transcription factors (AtSRs/CAMTAs), to induce the expression of immune-related genes [76] (pp. 795353). Furthermore, we validated the upregulation of CDPKs in response to *B. cinerea* infection, with a greater expression level observed in MT compared to AC (Figure 6a). Silencing analysis has demonstrated the positive regulatory roles of SlCDPK18 and SlCDPK10 in tomatoes’ resistance against non-host Xanthomonas oryzae pv. oryzae, while SlCRK6 positively regulates resistance against *Pst* DC3000 and *Sclerotinia sclerotiorum* [77] (pp. 661–676). Additionally, we investigated the expression of RBOH genes under *B. cinerea* invasion and observed a significant upregulation, particularly in MT (Figure 6a). It has been found that the activation of RBOH to generate reactive oxygen species (ROS) serves as an essential defense mechanism in plants. Recent studies have shown that C-terminal phosphorylation and ubiquitination can regulate the activity of AtRBOHD, highlighting the role of RBOH signaling in fine-tuning plant immunity [78] (pp. 1060–1062). In conclusion, our findings suggested that during the early stages of *B. cinerea* invasion in tomatoes, activation of the genes associated with Ca^2+^ signaling pathways may play a crucial role in recognizing and defending against pathogen invasion. Moreover, different tomato cultivars exhibited varying degrees of resistance, with MT showing a particularly robust response in this process compared to AC.

Notably, a significant number of DEGs are involved in Ca^2+^ signaling pathways, while a smaller subset was associated with phytohormone biosynthesis, stress responses, and PR proteins, among others (Figure 4 and Figure 5). Upon encountering pathogenic microorganisms, defense genes are immediately activated, leading to the rapid synthesis of ROS and immune-related proteins (such as PRRs) to counteract the invasion of pathogenic bacteria [79,80,81] (pp. 12990–12995, pp. 8577–8582, pp. 1487–1495). Our results revealed the upregulation of several PRRs involved in the recognition of PAMPs and initiation of immune responses in both tomato cultivars (Figure 6c). Specifically, genes such as CERK1, FLS2, EIX1, and BAK1 exhibited increased expression levels. Notably, the downstream gene MEKK1 displayed significantly higher upregulation in MT compared to AC. This upregulation indirectly regulated the activation of transcription factors, including members of the WRKY family, leading to the induction of immune-related genes such as NHO1 and PRs (Figure 5 and Figure 6). In summary, these findings emphasized the activation of PTI in tomato plants during the early stages of *B. cinerea* invasion, with a more robust response observed in the MT cultivar.

Meanwhile, this study demonstrated that the crucial role of ETI in tomato plants’ response to *B. cinerea* infection (Figure 5). We found that RPM1 and RPS2 homologous genes, involved in effector recognition and immune response initiation, were downregulated under *B. cinerea* induction in both tomato cultivars (Figure 4c). RAR1, SGT1, and HSP90 homologous genes were also suppressed in plant defense regulation (Figure 4c). In contrast, the upregulation of RIN4 and PBS1 homologous genes indicated their involvement in the immune response (Figure 4c). The EDS1-PAD4 complex may function as a second messenger in plant immunity [82] (pp. 495–499), and our results showed that these two genes were induced by *B. cinerea*, with particularly higher upregulation in MT (Figure 6b). Furthermore, our study highlighted the potential critical role of PIK1 in tomato plants’ immune response against *B. cinerea* (Figure 4c). Studies have demonstrated that Pto, a plant resistance protein, interacts with the Pti protein, which includes Pti1, Pti4, Pti5, and Pti6 [83] (p. 110702). These downstream molecules of the Pto-mediated ETI pathway promote the expression of disease-related genes and contribute to resistance against Pst DC3000 [84,85] (pp. 771–786, pp. 774229). In this study, we revealed significant upregulation of the Pti5 gene in both AC and MT, with a particularly pronounced upregulation in MT (Figure 6b), indicating that Pti5 played a more crucial role in the immune response against *B. cinerea* infection in tomatoes, especially in the MT dwarf cultivar. Meanwhile, Pti6 was induced to a higher level in AC than in MT, suggesting that Pti6 contributes to the defense mechanism in AC (Figure 4c). Researchers have found that tomato plants overexpressing Pti4/5/6 exhibited enhanced expression of pathological-related genes and resistance to the pathogenic bacteria Pst DC3000 [83,84,85,86] (p. 110702, pp. 771–786, p. 774229, pp. 1453–1457). Moreover, our study revealed two homologous genes of Pti1 with one significantly upregulated in MT and another in AC, indicating their potential role in plant immunity against pathogens (Figure 4c). In conclusion, our findings indicated that during the early stages of *B. cinerea* invasion in tomatoes, only a few genes in the ETI system were activated, emphasizing the selective gene activation within the tomato immune response to *B. cinerea*.

A robust body of evidence supports the idea that plants possess a dual-layered innate immune system known as PTI and ETI [87] (pp. 323–329). Recent studies suggest that PTI enhancement is an integral component of ETI during plant defense against pathogen invasion, and there exists a reciprocal promotion and regulation between PTI and ETI [31] (p. 102030). The upregulation of PTI amplifies the defense effects of ETI, while ETI activation can lead to PTI component upregulation. This interaction allows plants to effectively combat different types of pathogen invasions and provides comprehensive immune protection [31,32] (p. 102030, p. 1527). Studies have also shown that the activation of multiple NLRs (such as RPM1, RPS2, RPS5, RPS4, and RPP4) triggers the transcription and protein accumulation of several PRR signaling components in a PTI-independent manner, including BAK1, SOBIR1, BIK1/PBLs, RBOHD, and MPK3 [31,88] (p. 102030, pp. 485–497). Our study indicated that multiple PTI-related PRR genes, such as FLS2 and CERK1 were activated and significantly upregulated by *B. cinerea* invasion, particularly with a more pronounced upregulation in the MT (Figure 6c). Furthermore, several ETI-related NLR genes, including RPM1 and RPM2, were activated (Figure 4c). Additionally, RIN4, a gene involved in plant immune signal transduction and an interactor of RPM1, was significantly induced and upregulated upon *B. cinerea* invasion (Figure 4c). Moreover, genes associated with PTI, such as the marker genes PDF1.2, NHO1, and PR1 (Figure 6c), as well as other genes related to plant ETI, including pti1/4/5/6, RAR1, SGT1, HSP90, and EDS1, exhibited upregulated expression (Figure 4c), indicating their importance against *B. cinerea* invasion. Based on these findings, it is conceivable that these genes directly or indirectly participate in the process of PTI enhancing ETI. Collectively, these results indicated that during *B. cinerea* invasion, the activation of PTI in plants simultaneously enhanced the defense effects of ETI, albeit this process might be relatively limited in the early stages of *B. cinerea* infection in tomato plants.

Transcription factors regulate the expression of a lot of genes [89] (pp. 430–436). Previous studies have shown that seven TF families (such as WRKY, NAC, and MYB) play a crucial role in most plant immune processes against pathogens, including programmed cell death, activation of ROS, and hypersensitive cell death response [90,91,92,93,94,95,96,97] (pp. 1–13, pp. 1648–1655, pp. 932–947, pp. 85–106, p. 3737, pp. 1–29, p. 1064, pp. 335–351). In our study, we identified 53 TFs that regulate gene expression (Figure 7). In particular, during *B. cinerea* infection, a significant upregulation was observed in 20 WRKY, 14 MYB, and 11 NAC genes, particularly in the MT (Figure 6c and Figure 7a,b,d). Among the AUX/IAA TF genes, most exhibited differential expression and were predominantly upregulated (Figure 7c). These findings suggested that these transcription factors likely serve as positive regulators in the immune response of tomato against *B. cinerea* invasion.

Overall, we found there were a few differences in response to *B. cinerea* invasion between AC and MT cultivars and some crucial genes in regulating tomato immunity to *B. cinerea* invasion, particularly those confirmed by qRT-PCR, required functional validation experiments to reveal their regulatory roles.

## 4. Materials and Methods

The tomato plants were cultivated in a growth chamber at a stable temperature (22 °C) under a 12/12 h light/dark cycle, the humidity was 60–70% and irradiance was 125–145 uE·m^−2^·s^−1^. To establish an axenic growth environment, tomato seeds were sterilized by 10% NaClO solution for 2 min, followed by 4 rinses with sterile water. The sterilized seeds were then sown on sterile filter paper. For RNA sequencing (RNAseq) analysis, 4-week-old seedlings’ leaves were inoculated with 5 μL 1 × 10^5^ spores/mL *B. cinerea* conidia spores diluted in 10 mmol/L glucose and 10 mmol/L KH2PO4. Symptoms were photographed and analyzed at 12 h post inoculation (hpi), 14 hpi, 16 hpi, 18 hpi, 20 hpi, 22 hpi, 24 hpi, 26 hpi, 36 hpi, 42 hpi, and 48 hpi.

RNA was extracted using TRIzol reagent (Invitrogen, Waltham, MA, USA) and treated with RNase-free DNase (Solarbio, Beijing, China). First-strand cDNA synthesis was carried out using the PrimeScript RT reagent kit with gDNA Eraser according to the manufacturer’s instructions (Takara, Kusatsu, Japan). Transcriptome sequencing was conducted with three biological replicates, each of which comprised three technical replicates. In each technical replicate, samples were collected from three inoculated tomato plants for experimentation.

RNA sequencing (RNA-seq) of 12 libraries (CKAC01, CKAC02, CKAC03, AC01, AC02, AC03, CKMT01, CKMT02, CKMT03, MT01, MT02, and MT03; 01,02 and 03 presented one of three biological replicates) was performed by BioMarker (Beijing, China) using the PromethION sequencing platform from Oxford Nanopore Technologies (UK). Data quality control was carried out following the methods of CHEN et al. [98] (p. 997981), (1) which involved base calling of raw reads in the second-generation FAST5 format using the Guppy software from the MinKNOW2.2 software package [99] (p. 478172), (2) filtering of short and low-quality raw reads to obtain high-quality effective reads, and (3) identification of the primer sequences at both ends of effective reads according to the nanopore cDNA sequencing principle [100] (p. e38); reads with both primer sequences were deemed to be full-length transcript sequences. The sequencing data in this study were deposited in the NCBI Sequence Read Archive under the BioProject ID BMK190920-U840–0101. Principal component analysis (PCA) and Pearson correlation were performed based on the gene expression profiles obtained from FPKM (fragments per kilobase of exon per million fragments mapped) values of every million exon reads per kilobase base pairs of 12 samples in R. Annotation of the full-length transcripts was carried out using the Blast tool to compare against the NR [101] (pp. 1–8), Swissprot, KOG [102] (pp. 1–28), eggNOG [103] (pp. D231-D239), Pfam [104] (pp. D222–230), GO [105] (pp. 25–29), KEGG [106] (D277-D280), and COG [107] (pp. 33–36) databases to obtain corresponding functional and pathway annotation information.

Differential expression analysis of RNA sequencing data was performed using DESeq [41] (p. R106) for samples with biological replicates. In this transcriptome sequencing study, the counts per million (CPM) [108] (p. e91) were utilized as a measure of transcript or gene expression levels. CPM was calculated using the following formula: “reads mapped to transcript” represented the number of reads aligned to a specific transcript, while “total reads aligned in sample” represented the total number of fragments aligned to the reference transcriptome. To assess the dispersion of transcript expression levels within individual samples and facilitate visual comparisons of overall transcript expression levels across different samples, box plots were employed to display the distribution of CPM values. And during the detection of differentially expressed genes, a fold change ≥ 2 and a false discovery rate (FDR) < 0.01 were utilized as filtering criteria. The fold change represented the ratio of expression levels between two samples or groups, while the FDR was a corrected *p*-value obtained from adjusting the significance *p*-values of the differential expression. Due to the independent statistical hypothesis testing performed on lots of gene expression values in transcriptome sequencing differential expression analysis, there was a potential issue of false positives. Therefore, the widely recognized Benjamini–Hochberg correction method was employed to adjust the significance of *p*-values obtained from the original hypothesis testing, ultimately utilizing FDR as a key indicator for differential expression gene selection.

To identify the transcription factors (TFs) implicated in tomato leaf senescence under *B. cinerea* inoculation, we employed the iTAK software using its default settings [109] (pp. 1667–1670) to identify open reading frames (ORFs), which were then aligned to the TF protein domain using the HMM search database (version 3.0) [110] (p. e121). The resulting aligned sequences were annotated based on the available TF families in PlantTF database version 3.0 [111] (pp. D144–D149).

Quantitative real-time PCR: To confirm the reliability of our RNA-seq data, we selected 30 genes from four significant pathways identified by KEGG functional analysis and used quantitative real-time PCR (qRT-PCR) for validation. The qRT-PCR was performed with 3 technical replicates and 3 biological replicates using the SYBR Green Real-time PCR Master Mix (Takara, Japan) on a CFX96 Real-Time PCR Detection System. The samples consisted of CK (CKAC01, 02, 03, CKMT01, 02, 03 without *B. cinerea*) and samples (AC01, 02, 03, MT01, 02, 03 with *B. cinerea* treatment). The housekeeping gene Actin-Depolymerizing Factor 7 (*S. lycopersicum*) was used to standardize the relative expression. The delta-delta threshold cycle (2^−ΔΔCt^) method was employed to calculate the relative expression values for each of the 27 genes [112] (pp. 402–408). The primer sequences used in qRT-PCR are listed in Appendix A. The relative expression data were analyzed by ANOVA using SPSS (version 29, SPSS Inc., Chicago, IL, USA), and means were separated by DMRT at a 5% probability level.

Statistical analysis: The area of lesions was measured using ImageJ 1.43j software. All box plots and bar graphs were generated using GraphPad Prism v. 9.0 software. Statistical significance was determined by Student’s *t* test, one-way analysis of variance, or two-way analysis of variance followed by Tukey post hoc tests using IBM SPSS Statistics.

## 5. Conclusions

We found that the MT tomato cultivar exhibited strong resistance during the initial phases of *B. cinerea* infection but faster lesion expansion at later stages. Comparative analysis revealed differential gene expression profiles between MT and AC cultivars, with MT displaying a wider range of differentially expressed genes and extensive transcriptional reprogramming. The invasion of *B. cinerea* significantly impacted the regulation of genes related to carbon metabolism and photosynthesis in tomatoes. Differential gene expression and functional annotation analysis highlighted significant differences in proportions and annotations of DEGs between the CKMT-vs-MT and CKAC-vs-AC groups. Approximately one-third of DEGs in each group were associated with biological processes, particularly metabolism, immune system processes, and biological regulation and detoxification. Further KEGG analysis revealed that during the early stage of B. cinerea invasion, tomatoes depended heavily on Ca^2+^ channel-associated proteins, particularly in the MT cultivar. Additionally, partial activation of PTI, ETI, and SA signaling pathways occurred, involving genes such as FLS2, BAK1, CERK1, RPM, SGT1, and EDS1 associated with immune response. Furthermore, the WRKY, MYB, and NAC transcription factor families played crucial regulatory roles in tomatoes’ defense against *B. cinerea*. These findings provide valuable insights for enhancing tomatoes’ resistance against *B. cinerea* and pave the way for further research into potential targets and mechanisms.

## Figures and Tables

**Figure 1 plants-12-02965-f001:**
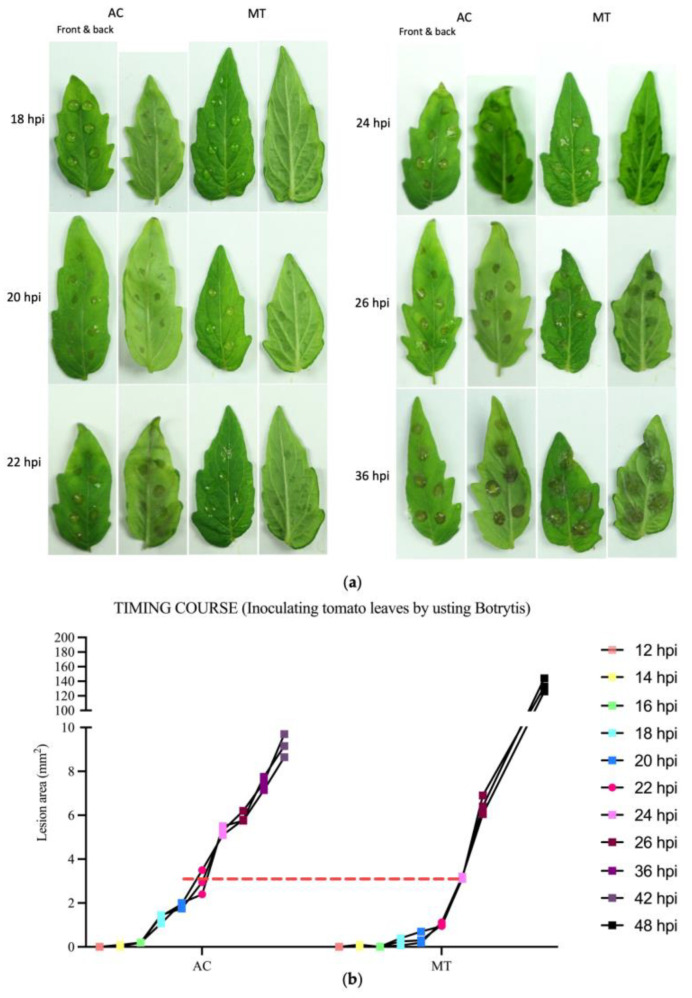
Time course of symptom development after inoculation with *B. cinerea* spores in vivo. (**a**) symptoms at 18 hpi, 20 hpi, 22 hpi, 24 hpi, 26 hpi, and 36 hpi; (**b**) lesion sizes at different time points. Four-week-old tomato leaves of two different cultivars, AC and MT, were inoculated with *B. cinerea* via the conidial inoculation approach (10^5^ conidia/mL, 5 μL) in vivo. Photographs were taken and lesions were measured at 12 hpi, 14 hpi, 16 hpi, 18 hpi, 20 hpi, 22 hpi, 24 hpi, 26 hpi, 36 hpi, 42 hpi, and 48 hpi.

**Figure 2 plants-12-02965-f002:**
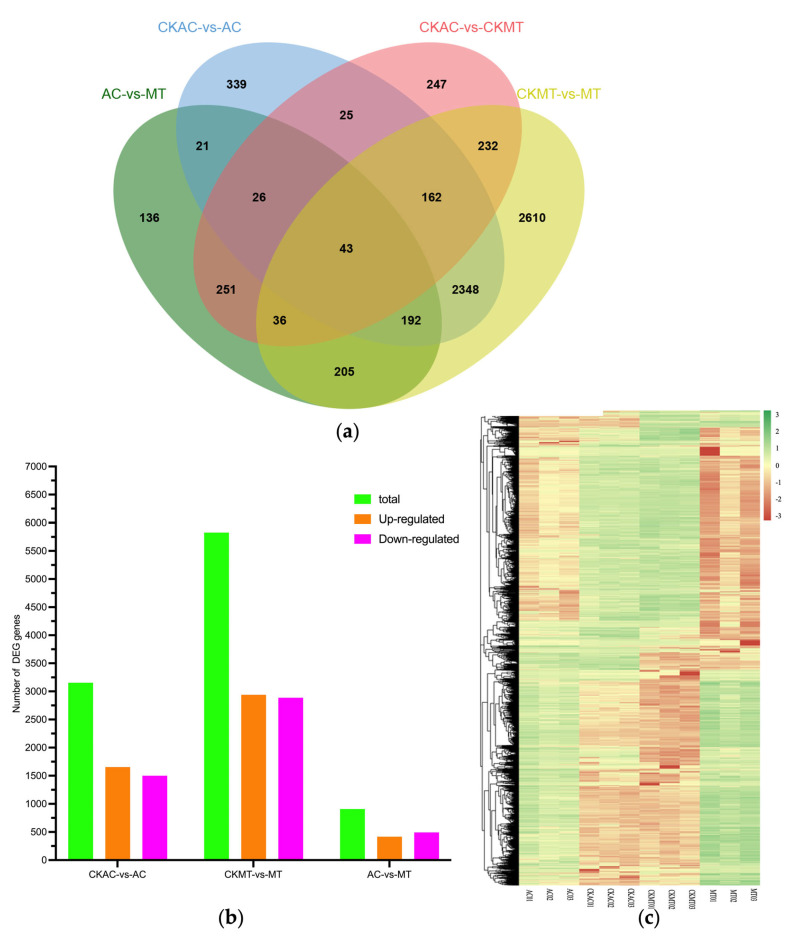
Analysis of differentially expressed genes (DEGs) in tomato cultivars AC and MT at 22 h and 24 h after inoculation with *B. cinerea* spores in vivo, respectively. (**a**) Venn diagram of DEGs in CKAC-vs-AC; CKAC-vs-CKMT; AC-vs-MT and CKMT-vs-MT. (**b**) The distribution of upregulated and downregulated genes of DEGs under different comparison combinations. (**c**) Heatmap illustrating the hierarchical clustering results for RNA-sequencing (RNA-seq). Each small square represented a gene and colors represented the expression levels. *n* = number of DEGs in each pairwise group; CKAC: AC without *B. cinerea*; AC: AC inoculated with *B. cinerea*; CKMT: MT without *B. cinerea*; MT: MT inoculated with *B. cinerea*.

**Figure 3 plants-12-02965-f003:**
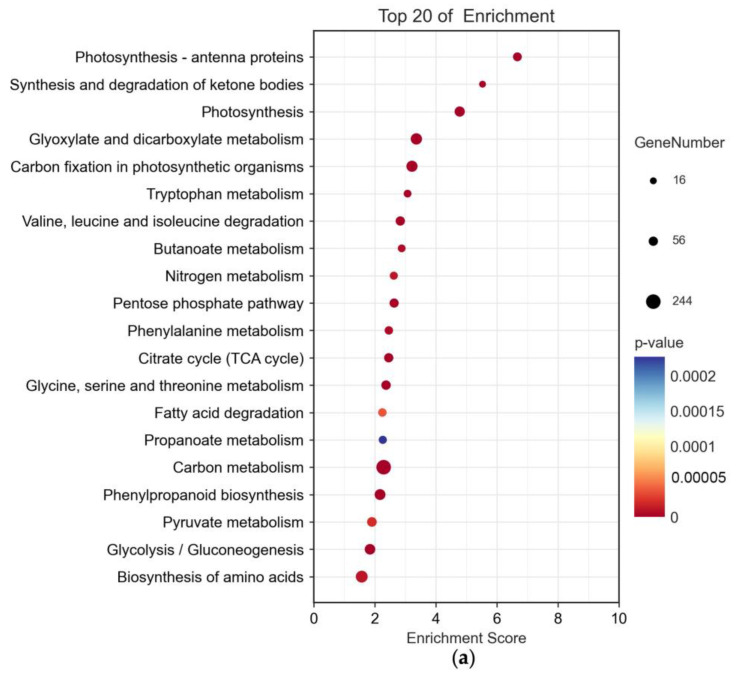
The top twenty enriched KEGG pathways based on the DEGs in two pairwise groups. (**a**) CKAC-vs-AC; (**b**) CKMT-vs-MT; (**c**) AC-vs-MT. Each circle in the figure represented a KEGG pathway, where the y-axis denoted the pathway name and the x-axis represented the enrichment factor, which was the ratio of the proportion of annotated genes in DEGs to the proportion of annotated genes in all genes for that pathway. A larger enrichment factor indicated a more significant enrichment level of DEGs in that pathway. The color of the circle represented the *p*-value, which was the value after multiple hypothesis testing correction. A smaller *p*-value indicated a more reliable significance of the enrichment of DEGs in that pathway. The size of the circle represented the number of enriched genes in the pathway, where a larger circle indicated more genes in the pathway.

**Figure 4 plants-12-02965-f004:**
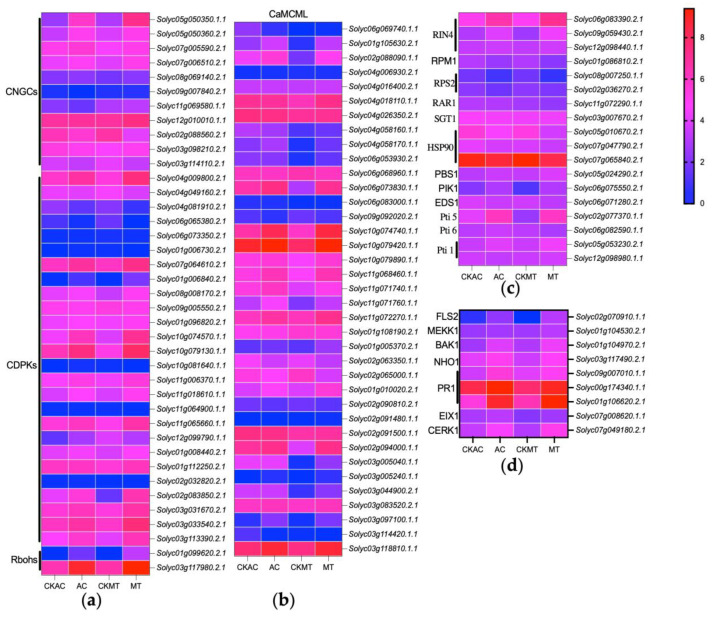
Expression profiling of critical DEGs involved in the plant immunity signal transduction pathway in Figure 5. (**a**) CDPKs, CNGCs and Rbohs; (**b**) CaMCMLs; (**c**) genes involved in the ETI pathway; (**d**) genes involved in the PTI pathway. Expression values of each gene were the log2-transformed expression values. Colors represented the expression levels.

**Figure 5 plants-12-02965-f005:**
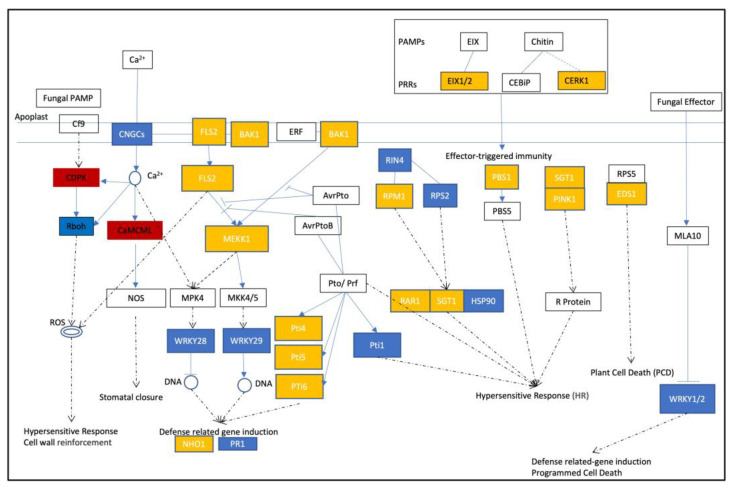
The schematic diagram of the plant hormone signal transduction pathway and its correlation with DEGs in response to *B. cinerea* treatment. The investigation focused on all DEGs. Within the diagram, key enzymes related to the pathway were represented by colored shaded rectangles. The colors within the diagram represent the number of DEGs linked to each enzyme. Red represents that the number of DEGs exceeds 10, indicating a notable and statistically significant differential expression pattern; blue represents that the number of DEGs ranges from 2 to 8, suggesting a moderate differential expression pattern; yellow represents that the number of DEGs is approximately 1 gene, implying a minor differential expression pattern.

**Figure 6 plants-12-02965-f006:**
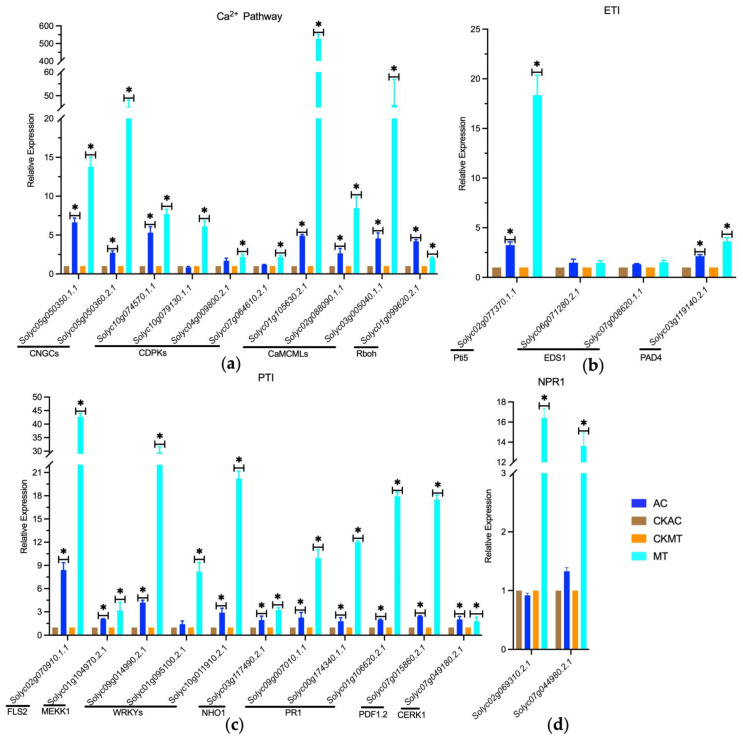
The expression patterns of validated genes involved in four prominent pathways associated with plant immunity by quantitative reverse transcription PCR (qRT-PCR). (**a**) Genes involved in calcium signaling pathways; (**b**) genes involved in ETI signaling pathways; (**c**) genes involved in the PTI signaling pathway; (**d**) genes involved in the SA signaling pathway. The data were calculated by the method of 2^−ΔΔCT^. The error bars indicate the standard deviation (STDEV) of the mean expression values. All data represent the mean of 3 biological replicates. Significant differences were denoted as follows ‘*’: (*p* ≤ 0.05).

**Figure 7 plants-12-02965-f007:**
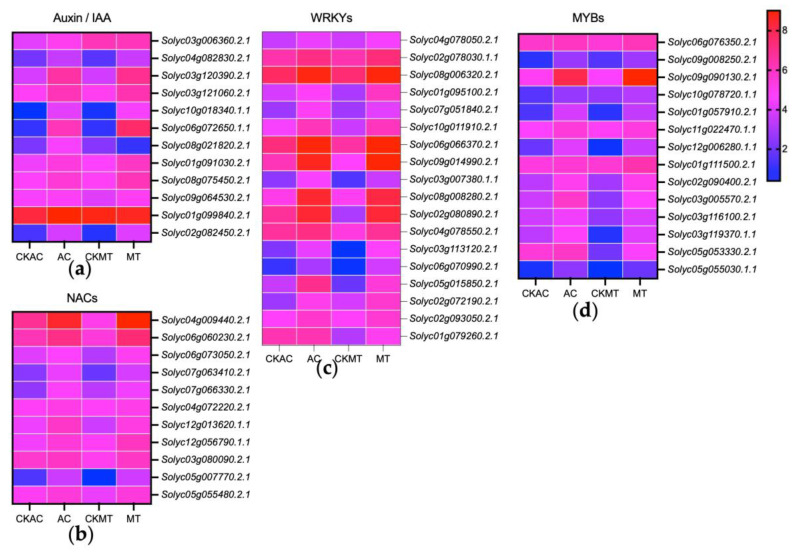
Expression profiling of differentially expressed major transcription factor families among the four groups (CKAC, AC, CKMT, MT) in leaves at 22 h (AC) and 24 h (MT) post-inoculation with *B. cinerea*. (**a**) AUX/IAA transcription factors regulating auxin signaling pathways; (**b**) NACs; (**c**) WRKYs; (**d**) MYBs. Expression values of each gene are the log2-transformed expression values. Colors represent the expression levels.

## Data Availability

All data generated or analyzed during this study are included in this article and its Appendix A. RNA-Seq data libraries were generated in this research. The sequences of tomatoes were downloaded from the tomato genome resources, accessible via the following link: https://phytozome-next.jgi.doe.gov/, accessed on 9 July 2023.

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
