# Peer review of "Unraveling the Molecular Mechanisms of Tomatoes’ Defense against Botrytis cinerea: Insights from Transcriptome Analysis of Micro-Tom and Regular Tomato Varieties"

_plants, 2023, doi:10.3390/plants12162965_

Round 1

Reviewer 1 Report

The manuscript addresses important point in the field of Tomato-botrytis cinerea interaction. There are few reports from Harel lab and other, those need to cited in the article. The article needs few minor english revision such as

Line-103, response instead of succeptibility.

The article needs few minor english revision such as

Line-103, response instead of succeptibility.

Reviewer 2 Report

I am writing in regard to manuscript # 2527061 entitled "Unraveling the Molecular Mechanisms of Tomato's Defense Against 2 Botrytis cinerea: Insights from Transcriptome Analysis of Micro-Tom and 3 Regular Tomato Varieties ", which you submitted to Plants.  Firstly, I congratulate the authors on the considerable amount of work this study represents. The work described in this well written manuscript is thorough and original and it would be a valuable addition to the grey mold disease control literature. 

 The study involves a comprehensive reflection of molecular mechanisms driving plant responds to Botrytis cinerea by pathway analysis and functional annotation to find differences in gene expression.

On the other hand, the overall formal quality of the manuscript is acceptable, although some parts might be easily improved. Several conclusions were drawn from this research, supporting the notion of beneficial effect plant resistance and the potential use of biocontrol agents with plant induced responds mode of actions.

No comments.

Reviewer 3 Report

The paper studied the molecular defense mechanism of tomato MT and AC against B. cinerea. The overall is good. However, some suggestions need to be revised:

1. The methods are not clear enough. The authors do not mention how many plants were used to collect samples and how many experiments were performed for transcriptome analysis?

2. Figure 5 is not in a good quality

3. Figure 4 can be put in the supporting files

The language is good and minor errors should be corrected all through the paper.
